# Designing Micro Bulge Structure with Uniform PS Microspheres for Boosted Dielectric Hydrophobic Blend Films

**DOI:** 10.3390/polym12030574

**Published:** 2020-03-04

**Authors:** Zhicai Zhu, Liqin Yao, Qilong Zhang, Hui Yang

**Affiliations:** School of Materials Science and Engineering, State Key Lab Silicon Mat, Zhejiang University, Hangzhou 310027, China; 21726096@zju.edu.cn (Z.Z.); 21926063@zju.edu.cn (L.Y.); yanghui@zju.edu.cn (H.Y.)

**Keywords:** dielectric properties, hydrophobicity, micro bulge structure, interfacial polarization

## Abstract

In this paper, homogeneous polystyrene (PS) microspheres with controllable sizes of 40 nm, 80 nm, and 120 nm were synthesized by controlling the temperature of solvothermal method. In order to explore the effect of PS microspheres on dielectric-hydrophobic properties of the composite films, the composite films containing polystyrene, Polydimethylsiloxane, and P(VDF-TrFE) with high dielectric and hydrophobicity were successfully prepared by a simple and feasible solution blending method. The dielectric constant and hydrophobicity of composite films were boosted by increasing the mass fraction of PS content and decreasing the size of PS due to the enhanced interfacial polarization and the uniform surface micro bulge structure. Meanwhile, the composite films maintain a low loss tangent. Typically, the dielectric constant with 5 wt.% 40 nm PS reached to 29 at 100Hz, which is 4 times that of PDMS/P(VDF-TrFE) (mass ratio: 2/3). Otherwise, the largest the contact angle of 126° in the same composition was remarkably larger than the pure PDMS/P(VDF-TrFE) (110°). These improved properties have more potential applications in the electric wetting devices.

## 1. Introduction

Electrowetting on dielectrics (EWOD) is a technical means to control and treat small droplets by means of electrowetting phenomenon. The control means mainly depend on the influence of applied voltage on the interface tension to control the wettability of solid-liquid surface. EWOD has been widely applied in biochip, optical device, display, on-chip laboratory, and other fields [1,2]. The Young–Lippmann equation describes the principle of EWOD-based voltage dependence of polar liquid, from which we can concluded that the ratio of the thickness of the dielectric layer to the dielectric constant determines the threshold voltage. Thinning the dielectric layer or enhancing the dielectric constant of the insulating layer can effectively reduce the threshold voltage, but the thin dielectric layer is easy to break down when driven by voltage. Therefore, the improvement of dielectric constant and hydrophobicity of dielectric layer plays a decisive role in the application of low-voltage electrowetting devices.

So far, there has been extensive research on various inorganic or polymer materials that can be applied to the dielectric materials of EWOD. Polymer materials like polydimethylsiloxane, poly-*p*-xylene, SU-8 UV photoresist, polyimide, and Teflon AC have exceptional hydrophobic properties [3,4,5]. However, the low dielectric constant and high driving voltage limit their wide application in EWOD. Corresponding inorganic materials like SiO_2_, Ta_2_O_5_, Al_2_O_3_, Si_3_N_4_ have high dielectric constants and are also hydrophilic, and the preparation methods of inorganic materials such as chemical vapor deposition (CVD), atomic layer deposition (ALD), and ion sputtering are more complex, and the experimental conditions are usually at high temperature and high pressure [6,7]. In order to make full use of the advantages of inorganic and polymer dielectric materials, some researchers have proposed a two-layer structure with a composite of an organic hydrophobic layer on the surface and an inorganic layer with a high dielectric constant on the bottom [6,8]. However, the manufacturing process of this structural design is more complicated. Moreover, due to the tremendous difference in physical and chemical properties between the organic hydrophobic layer and the inorganic dielectric layer, the bonding strength between the two layers is weak, which will further cause interlayer mismatch, interlayer gaps and even separation.

In recent years, in order to change the disadvantage of low dielectric constant of polymer materials, more and more attention has been paid to the composite materials with inorganic fillers embedded in a hydrophobic organic matrix. Organic/inorganic nanocomposites are a new type of composite materials, which are formed in the range of micron or even nanometer by chemical bonding with organic matrix and inorganic particles as raw materials. This kind of material not only has the characteristics of inorganic filler, but also remains polymer characteristics, at the same time, it also has the properties that neither of them has, so it has very good application value [9,10]. For instance, Mao et al. [11] added BaTiO_3_ with dielectric constant over 1000 to PVDF to investigate the effect of filler particle size on dielectric properties. The results showed that the volume fraction of filler with particle size of 80–100 nm is 60%, and the dielectric constant is as high as 93 (1.14 kHz). Zhang et al. [12] dispersed BaTiO_3_ particles in TiO_2_ nanowires by electrospinning, and the dielectric constant of TiO_2_ was 40 between BaTiO_3_ and PVDF. Meanwhile, the good dispersion of TiO_2_ nanowires directly improved the dispersion of BaTiO_3_, and the breakdown strength was as high as 600 mV/m when the amount of powder was only 3%. In addition to the research on the composite of such insulating particles and polymer system, the conductor or semiconductor particles are also combined with polymer system, and the dielectric constant of the composite material will increase suddenly after adding conductive materials, which is called deep current phenomenon. When the filler content exceeds this value, the dielectric changes from insulator to conductor, and the critical value of the change is called the percolation threshold. Near the percolation threshold, the distance between conductive particles is relatively close, which can be regarded as a large number of parallel micro capacitors, which makes the dielectric constant suddenly increase, and also leads to a large loss and conductivity. Xu et al. [13] composite Al particles with epoxy resin, the surface of Al will be passivated spontaneously to form a dense layer of Al_2_O_3_, so that the Al particles will not directly contact with each other, resulting in the composites with high dielectric constant and low loss. However, high dielectric films usually need inorganic fillers to reach high volume fraction, which inevitably aggravates the homogeneity, flexibility, and processing ability of composites.

In this paper, a self-made small-sized organic polystyrene (PS) was introduced to prepare a uniform high dielectric hydrophobic film, and a new method for preparing a composite film was proposed. In the previous work, we selected PDMS elastomer with excellent hydrophobicity and P(VDF-TrFE) to provide high dielectric constant to composite them as polymer matrix according to the mass ratio of 2:3. The surface of PDMS/P(VDF-TrFE) organic matrix film is smooth and compact, the dielectric constant is 7 at 100 Hz, and the initial hydrophobic angle is 110°. In order to further improve the dielectric constant and hydrophobicity of the composite films, PS microspheres with small diameter (40–120 nm) were synthesized by solvothermal method and blended with PDMS and P(VDF-TrFE) to form a micro bulge structure to improve the hydrophobic properties, and enhance the interface polarization to improve the dielectric properties. The composite films with improved properties have more potential applications in the electric wetting devices. In addition, PS nanoparticles with negative charge can also be used for calibration of microfluidic devices because of their uniform and controllable particle size. In microfluidic applications such as flow cytometry, cell sorting, and imaging, it is a key capability to three-dimensionally focus randomly distributed particles into a single flow [14,15].

## 2. Experimental

### 2.1. Synthesis of PS Microspheres

PS microspheres with small particle size were prepared by solvothermal method, and the particle size of PS microspheres at different temperatures was explored. First, 1.5 mL styrene(C_8_H_8_) (Sinopharm Chemical Reagent Co. Ltd, Shanghai, China) was dispersed in 20 mL deionized water and 20 mL acetone (CH_3_COCH_3_) (Aladdin Industrial Corp., Shanghai, China) mixed solution. Second, 0.02g 2,2’-Azobis(2-methylpropionamide) dihydrochloride (AIBA) (Aladdin Industrial Corp., Shanghai, China) was added, stirred for 10 min, and solvent heated at 90 °C for 6 h (other particle size can change the solvothermal temperature, 80 nm PS can be obtained at 70 °C, 120 nm PS can be obtained at 50 °C). Last, PS microspheres with a particle size of 40 nm were obtained by ethanol washing, centrifugation, and drying at 60 °C.

### 2.2. Fabrication of the Composite Films

First, various mass fraction and different size of PS microspheres were dissolved with 0.6 g P(VDF-TrFE) (50:50 mol%, provided from Wuhan Cymenes Technology Co. Ltd., Wuhan, China) in *N*,*N*-dimethylformamide (DMF) (Aladdin Industrial Corporation, Shanghai, China) under vigorous stirring. Meanwhile, 0.4 g Polydimethylsiloxane (PDMS) (Dow Corning Corp., Auburn, MI, USA) was dissolved in 2 mL isopropanol (C_3_H_8_O) by mechanical stirring. After 24 h, the two solutions were mixed together, and 0.04 g silicone elastomer curing agent (Dow Corning Corp., Auburn, MI, USA) was added, before being stirred in water bath for 24 h. After the mixed solution was uniformly dispersed, it was vacuumed for 1 h and spin coating to form a film, the remaining solution is made into a casting film. Finally, the composite film was dried under normal temperature for a whole night to prevent the formation of holes due to rapid evaporation rate of solvent, and then the film was dried at 80 °C for 24 h.

### 2.3. Characterization

Scanning electron microscopy (FESEM, SU-8010, Hitachi, Tokyo, Japan) was used to investigate the surface and cross-section morphology of the composites. Transmission electron microscopy (TEM, HT-7700, Hitachi Ltd., Tokyo, Japan) was used to characterize the diameter and dispersion of PS microspheres. The crystallization properties of the composite films were tested by X-ray diffractometer (XRD, EMPYREAN, PANalytical Co., Almelo, The Netherlands, Cu-Kα, 2θ = 10–80°). The characteristic absorption peaks of the chemical functional groups in the spectral range of 3500–500 cm^−1^ were performed by Fourier transform infrared spectrometer (FTIR, Nicolet 5700, Thermo Fisher, Waltham, WA, USA). The thermal stability of composites was evaluated by thermogravimetry (TGA, Q500, TA Instruments, New Castle, DE, USA) at 60–200 °C (10 °C/min). The dielectric properties of the composite films were obtained by 4294A Precision Impedance Analyzer (Agilent, Palo Alto, CA, USA) at 10^2^–10^6^ Hz. The initial contact angle (CA) of the sample was measured by OCA20 video contact angle tester (DataPhysics Co., Stuttgart, Germany). The surface roughness of the composite films was analyzed by using the multi-mode scanning probe microscope (Multi Mode, VEECO, Plainview, NY, USA).

## 3. Results and Discussion

Styrene is a hydrophobic monomer and the nucleation process “micelle nucleation”, solvothermal method is used to introduce the reaction conditions of high temperature and pressure in the closed system. The free radicals of micellar oligomers can be subdivided into smaller free micelles further under high temperature and pressure conditions, and smaller PS microspheres can be formed after nucleation. The particle size of PS microspheres can be regulated by controlling reaction temperature. As shown in Figure 1, PS microspheres with uniform particle size distribution of 40 nm, 80 nm, and 120 nm were successfully prepared by controlling reaction temperature of 90 °C, 70 °C, 50 °C, and reaction time of 6 h. By zeta potential measurement, the external potential of the ball was −7.62 mV in Figure 2, which was preparation for the subsequent test and characterization.

SEM was used to study the surface and cross-section morphology of the composite films, as shown in Figure 3. The surface of PDMS/P(VDF-TrFE) organic matrix film was dense without holes, and the roughness of the material surface had an important influence on the initial contact angle. However, after the incorporation of small particle size PS, the surface had obvious micro bulge structure, which increased with the increase of mass fraction. Combined with the surface roughness value in Figure 4, the surface roughness increased with the increase of mass fraction. Compared with PS spheres with different diameters in the same mass fraction, the micro bulge structure increased with the increase of the size of PS microspheres. This can be mainly found from the cross-sectional views of different particle size, in that the smaller the particle size, the better mechanical compatibility with the organic matrix, the less obvious the bulge structure on the surface and inside, so the roughness increases with the increase of the size of PS microspheres.

The surface roughness of PS/PDMS/P(VDF-TrFE) composite films with small particle size was measured by SPM. As shown in Figure 4, the roughness of pure PDMS/P(VDF-TrFE) organic matrix film was 4.4 nm, but with the addition of PS spheres, the roughness increased gradually. The maximum roughness of the blend film was 5 wt.% 120 nm PS component, and the roughness was 129 nm. The roughness value of 5 wt.% 40 nm PS blend film with better dielectric and hydrophobic properties was 33 nm, which provided the possibility for the potential application of our blend films in electrowetting devices.

Figure 5 illustrates the XRD patterns of the composite films with different diameters and various mass fraction of PS microspheres. In the P(VDF-TrFE) pattern, there was a clear and sharp diffraction peak of about 19.2°, which corresponded to the (110) and (200) crystal faces of the ferroelectric β phase of P(VDF-TrFE). It can be concluded from X-ray diffraction that the crystallizing peak intensity of β phase decreased with the addition of amorphous PDMS and PS, but the crystallizing peak intensity of the blend films increased with the increase of the mass fraction of PS microspheres when the particle size of PS microspheres was 40 nm. This is mainly due to the negative charge on the surface of PS microspheres, and the existence of the negative charge will interact with the –CH_2_ group in P(VDF-TrFE), resulting in the formation of polar β phase, thus increasing the intensity of crystallization peak [16]. However, compared with PS-doped blend films with different particle sizes under the same mass fraction, the crystalline peak intensity decreased with the increase of PS particle size, which may be caused by the smaller the particle size; the better the mechanical compatibility of the PS microspheres and the organic matrix is, the stronger the binding ability is. Therefore, more PS microspheres dispersed in the organic matrix. Further, the larger the particle size is, the worse the solubility with the organic matrix film is, the worse the ability to induce β phase formation is, and the lower the crystallization peak strength is.

As shown in Figure 6, we characterized the FTIR transmission spectra of PS blend films with different particle sizes and mass fractions to analyze the crystal structure and organic functional groups of the composite films. According to results in the figure, the absorption peak at 1078 cm^−1^ owed to the C=C stretching of β phase [17,18], and the peak intensity became stronger with the increase of PS microspheres mass fraction. This is because the PS surface is negatively charged, and the existence of negative charge will interact with –CH_2_ group in P(VDF-TrFE) to induce the formation of polar β phase, so the C=C stretching of β phase is enhanced [16]. However, when the mass fraction remained unchanged and the particle size of PS microspheres increased, the absorption peak strength here gradually weakened, which is consistent with the rule in the XRD pattern. The main reason is that the smaller the particle size is, the better mechanical compatibility of PS microspheres and organic matrix is, and the stronger the binding capacity is. Therefore, when the particle size increased, the ability to induce β phase formation weakened, and the C=C tensile vibration peak of β phase decreased correspondingly. The absorption band at 699 cm^−1^ is caused by the deformation vibration band between the carbon atom on the benzene ring of PS microsphere and the hydrogen atom linked to it [19]. There were obvious absorption peaks at 790 cm^−1^, 1257 cm^−1^, 1413 cm^−1^, and 2966 cm^−1^, which were the characteristics of Si–CH_3_ tensile vibration of PDMS [20,21]. Comparing the infrared spectrum of the blend films with PS and the spectrum of the organic matrix blend film without PS, we find that there were no other characteristic absorption band. Therefore, it can be concluded that the chemical functional groups of the blend films will not be changed and the crystallization performance of the ferroelectric phase of the blend films will only be affected by the addition of PS, regardless of the mass fraction.

Figure 7a,b are differential scanning calorimetric curves of blend films with different mass fraction and various diameters of PS. Because the glass transition temperature (*T*_g_) of blend films were not obvious, we used the differential derivative curve of modulation melting curve in Figure 7c to analyze the location of *T*_g_. As shown in Figure 7a, small size PS/PDMS/P(VDF-TrFE) composite films had peaks at about 134 °C and weak peaks at about 56 °C, which were crystallization peaks and phase transition points from paraelectric to ferroelectric. Under the temperature of ferroelectric paraelectric transition (F-P), the crystal had a long sequence of all trans molecular structure. More and more gauche conformations were introduced above the ferroelectric paraelectric transition temperature. Therefore, the polarity of the crystal region is reduced. The results show that the curie temperature and crystallization peak temperature of pure PDMS/P(VDF-TrFE) organic matrix composite and small size PS/PDMS/P(VDF-TrFE) composite were basically the same, which shows that the incorporation of PS had no significant effect on the F-P phase transition and crystallization temperature of the composite films [22,23]. Figure 7b shows the melting curve of the blend film with different mass fraction. The peak at about 163 °C was the melting peak. With the addition of different particle size and different mass fraction of PS, the melting temperatures are basically the same, indicating that the addition of PS has no obvious effect on the melting temperature of the composite films. According to the literature [9,22,24,25], the glass transition temperature (*T*_g_) was about −20 °C, but because the temperature signal of the transition point was weak, we used the same differential scanning calorimeter to analyze the modulation melting curve and get the curve of temperature derivative of the modulation melting curve in Figure 7c. The glass transition temperature (*T*_g_) was about −20 °C. The *T*_g_ values of the blend films with different mass fractions are shown in Table 1. Compared with the *T*_g_ of pure PDMS/P(VDF-TrFE) organic matrix blend film, the *T*_g_ value first increased and then decreased with the increase of mass fraction of PS. This is mainly because the *T*_g_ of PS was around 110 °C, and PS made the *T*_g_ of blend films move towards the higher temperature direction. As shown in Figure 5, the ratio of amorphous composition to crystalline decreased with increasing the PS content, which leads to the lower temperature (*T*_g_) of the amorphous composition changing from glassy state to high elastic state. For the samples with the same PS doping content, the *T*_g_ value increased with increasing the particle size. This is mainly attributed to the fact that the crystallization peak strength decreased with increasing the particle size of PS microspheres, which is proved by the XRD of Figure 5.

The initial contact angle is one of the important bases to judge the quality of a dielectric wetting material. As shown in Figure 8, we used the video contact angle measuring instrument to test and characterize the influence of different particle size and mass fraction on the initial contact angle of the composite film. It can be seen from Figure 8 that the initial contact angle of the blend film increased gradually with the increase of mass fraction. This phenomenon can be explained by Wenzel wetting theory [26,27,28,29,30,31]. According to the Wenzel theory, by adding PS microspheres, we built a micro convex structure on the surface of the composite film, thus increasing the surface roughness of the composite film, increasing solid–liquid contact area and enhancing hydrophobic property. When the content of 40 nm PS microspheres was 5 wt.%, the initial contact angle of the blend film reached 126°, while the initial contact angle of PDMS/P(VDF-TrFE) organic matrix composite was only 110°. When the amount of filler remained unchanged, the initial contact angle of the blend film decreased with the increase of the particle size of PS microspheres. This may be mainly because the smaller the particle size is, the better mechanical compatibility of PS microspheres and organic matrix is, and the stronger the binding capacity is, so that the initial contact angle of the blend film decreases with the particle size. However, the initial contact angle of the composite films with 5 wt.% doping can reach more than 120° regardless of the particle size, which provides the possibility for the application of the blend film in the electrowetting device.

The dielectric properties of different blend films are shown in Figure 9. In different frequency ranges from low frequency to high frequency, the contribution of medium polarization was different. The polarization in medium can be divided into space charge, electron, ion, and dipole polarization. In the low-frequency range (10^2^–10^6^ Hz), space charge polarization had an important influence on the dielectric properties. Ion polarization and electron polarization play a main role to affect the dielectric properties when the frequency increased above 10^6^ Hz [32,33,34,35]. The main contribution of blend films to the dielectric constant was the interface polarization, which is a kind of space charge polarization. The Maxwell–Wagner–Silars (MWS) theory explains that the effect of interface polarization enhances the dielectric constant of materials in the low-frequency region. In the two heterogeneous materials, due to the different conductivity, the charge will gather on the two-phase interface with different relaxation time when the small current formed by electromagnetic wave passes through the two-phase interface, which will greatly increase the dielectric constant of the material. The MWS polarization of the composite films includes MWS_PDMS-P(VDF-TrFE)_ and MWS_PS-PDMS/P(VDF-TrFE)_, in which MWS_PS-PDMS/P(VDF-TrFE)_ has an important influence. As a result, the dielectric constant of the composite film increased with the increase of PS mass fraction, and reached the maximum value at 5 wt.% 40 nm PS (~29@100 Hz), which was almost 4 times that of pure PDMS/P(VDF-TrFE) composite film. However, when the mass fraction was the same and the particle size of PS increased, the dielectric constant of 5 wt.% was 120 nm PS blend film (at 100 Hz, ~23). This is mainly because the smaller the particle size of PS is, the better the binding ability with the organic matrix is, the more obvious the polarization effect of MWS_PS-PDMS/P (VDF-TrFE)_ interface is, and the dielectric constant decreases with the increase of particle size at the same frequency and the same mass fraction.

Figure 9b displays the dielectric loss of composites with different mass fraction and particle size PS, and the corresponding dielectric loss also shows frequency dependence. The results show that the dielectric loss of the blend films was almost independent of the content and size of PS microspheres at a lower frequency (below 10^4^ Hz), but the dielectric loss of PDMS/P(VDF-TrFE) organic matrix blend films without PS doping began to increase at 10^4^ Hz. When the frequency was further increased, the dielectric loss of the blend film increased, which was mainly caused by the interface polarization and dielectric relaxation in the PVDF matrix [36,37,38]. The results show that we can improve the dielectric properties by reducing the mass fraction and the size of PS spheres. This is mainly due to the increase of the solubility of PS with organic matrix and the increase of the binding capacity after the decrease of PS particle size, resulting in the obvious interface polarization effect. At the same time, the existence of PS microspheres in media limits the mobility of organic molecular chains, which makes the dielectric loss of the composite film with 40 nm PS at high frequency equal to or even smaller than the pure PDMS/P(VDF-TrFE) film.

Figure 9c shows the conductivity of the blend film at frequencies of 10^2^ to 10^6^ Hz. As shown in the figure, the conductivity of the composite film increased with the increase of mass fraction, which may be due to a small leakage current between the interface polarization charges caused by adding PS and a small number of defects in the blend film after adding fillers. However, the conductivity of the composite films can keep a very low value (less than 10^−7^ s/cm) below 10^4^ Hz, indicating that the composites maintain excellent insulation performance. When the frequency was further increased, the conductivity of the blend film increased rapidly, which can be explained as the excitation of electric charge [39].

## 4. Conclusions

In conclusion, the uniform PS microspheres with the different size were synthesized by solvothermal method. Blend films with homogeneously dispersed PS microspheres in PDMS/P(VDF-TrFE) (mass ratio: 2/3) were prepared. The influences of the mass fraction and the size of PS microspheres on the melting and crystallization characteristics, the surface morphology, and the dielectric hydrophobicity of the composite films were systematically analyzed. The dielectric constant and hydrophobicity of the composite film boosted significantly with increasing the PS content and decreasing the size of PS, which is attributed to the enhanced interface polarization effect and the formation of uniform micro bulge structure. Typically, the dielectric constant with 5 wt.% 40 nm PS reached 29 at 100 Hz, which is 4 times of PDMS/P(VDF-TrFE) (mass ratio: 2/3). Meanwhile, the largest the contact angle of 126° in the same composition was remarkably larger than the pure PDMS/P(VDF-TrFE) (110°).

## Figures and Tables

**Figure 1 polymers-12-00574-f001:**
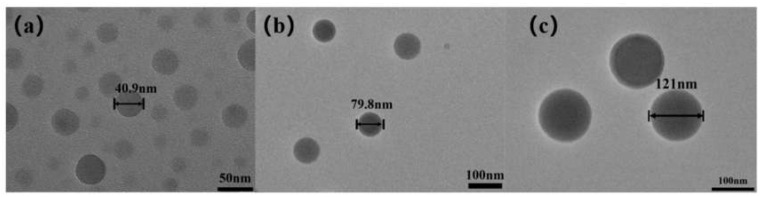
TEM of polystyrene (PS) microspheres at various reaction temperature:(**a**) 90 °C, (**b**) 70 °C, (**c**) 50 °C.

**Figure 2 polymers-12-00574-f002:**
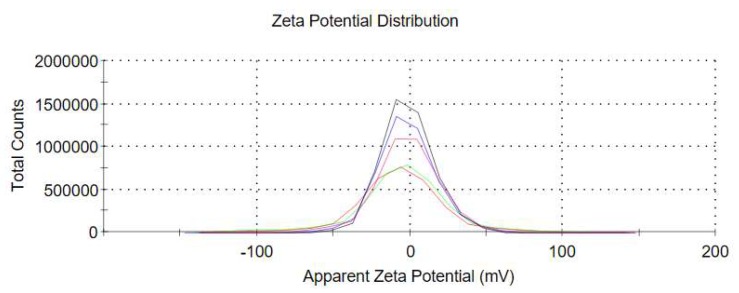
Zeta potential of 40 nm PS spheres in *N*,*N*-dimethylformamide (DMF) at room temperature.

**Figure 3 polymers-12-00574-f003:**
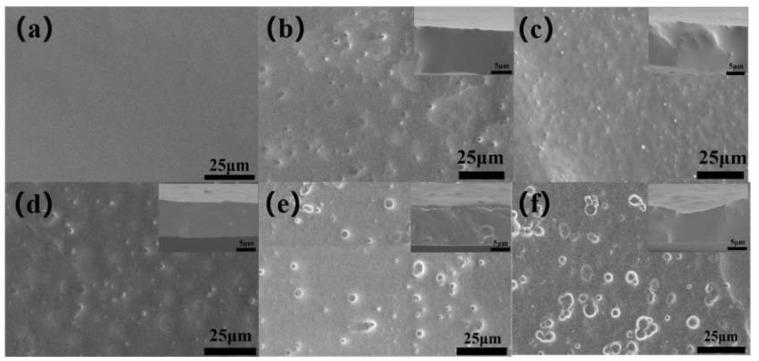
SEM of the surface and cross-section of the composite films with (**a**) 0 wt.% PS, (**b**) 1 wt.% 40 nm PS, (**c**) 3 wt.% 40 nm PS, (**d**) 5 wt.% 40 nm PS, (**e**) 5 wt.% 80 nm PS, (**f**) 5 wt.% 120 nm PS.

**Figure 4 polymers-12-00574-f004:**
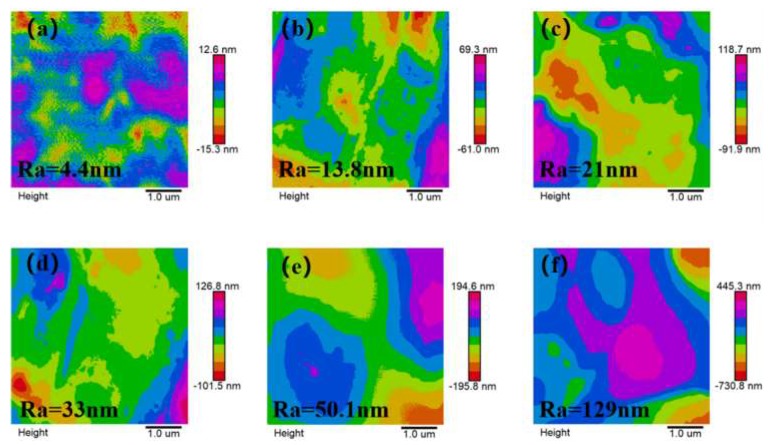
Surface roughness of blend films with different mass fraction (**a**) 0 wt.% PS, (**b**) 1 wt.% 40 nm PS, (**c**) 3 wt.% 40 nm PS, (**d**) 5 wt.% 40 nm PS, (**e**) 5 wt.% 80 nm PS, (**f**) 5 wt.% 120 nm PS.

**Figure 5 polymers-12-00574-f005:**
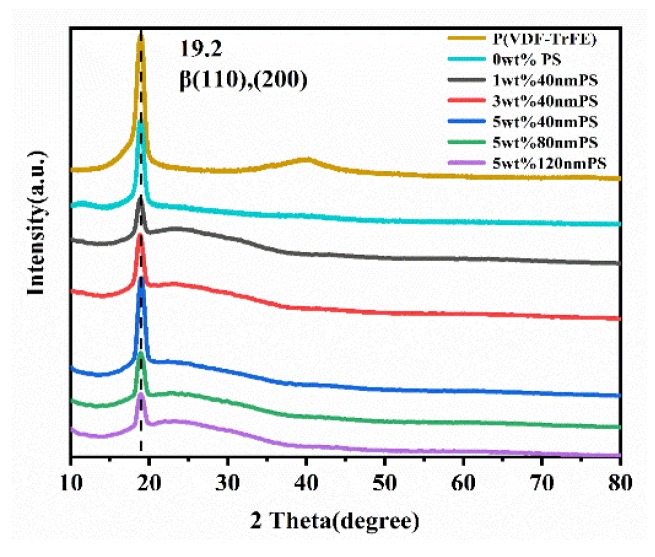
XRD patterns of blend films.

**Figure 6 polymers-12-00574-f006:**
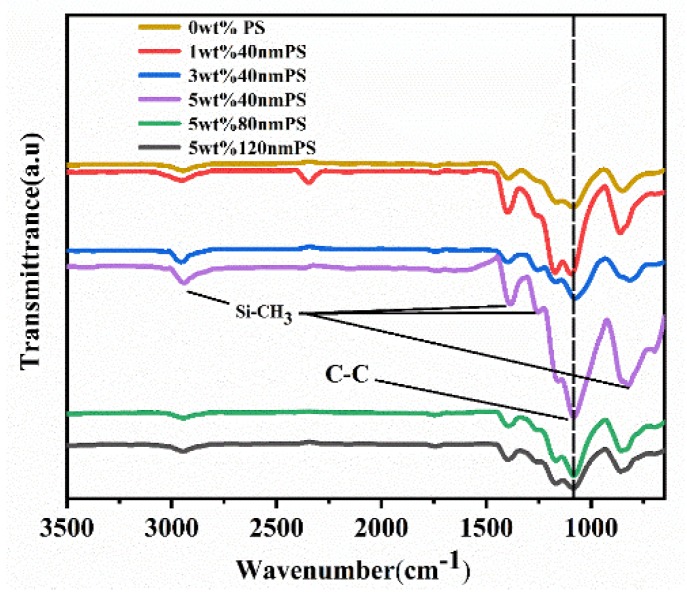
FT-IR spectra of blend films.

**Figure 7 polymers-12-00574-f007:**
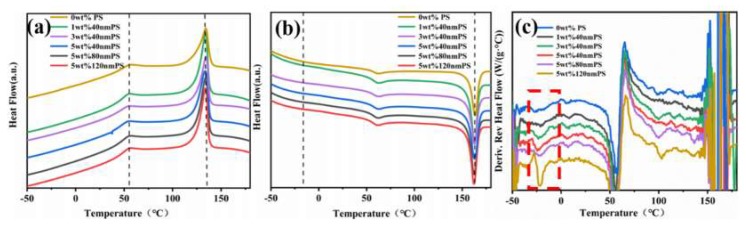
(**a**) Crystallization curve, (**b**) melting curve, and (**c**) derivative curve of modulation melting curve for temperature of blend films with different mass fraction.

**Figure 8 polymers-12-00574-f008:**
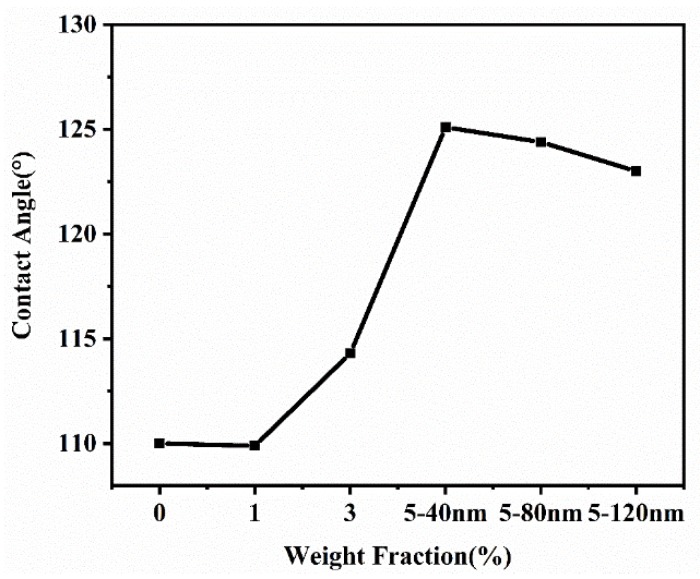
Hydrophobic angle of blend film with different PS mass fraction.

**Figure 9 polymers-12-00574-f009:**
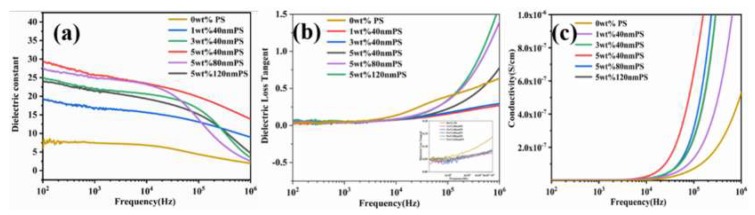
Dielectric properties of blend films with different PS mass fraction: (**a**) Dielectric constant, (**b**) dielectric loss, and (**c**) electric conductivity.

**Table 1 polymers-12-00574-t001:** The value of *T*_g_ of various mass fraction of the composite films.

	0 wt.% PS	1 wt.% 40 nm PS	3 wt.% 40 nm PS	5 wt.% 40 nm PS	5 wt.% 80 nm PS	5 wt.% 120 nm PS
*T*_g_/°C	−27.03	−21.28	−23.09	−24.30	−22.49	−21.88

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
