# Peer review of "Designing Micro Bulge Structure with Uniform PS Microspheres for Boosted Dielectric Hydrophobic Blend Films"

_polymers, 2020, doi:10.3390/polym12030574_

Round 1

Reviewer 1 Report

Dear authors,

This manuscript successfully demonstrated  a simple and feasible solution blending method to prepare the composite films containing polystyrene, Polydimethylsiloxane, P(VDF-TrFE) with high dielectric and hydrophobicty. The dielectric constant and hydrophobicity of composite films were boosted by increasing the mass fraction of PS content and decreasing the size of PS due to the enhanced interfacial polarization and the uniform surface micro bulge structure.

Due to the very broad application in addition to EWOD, several microfluidic devices also use PS beads for device calibration. With this kind of new method, authors are recommended to develop new types of polymer beads for such kind to application. It would be great if the authors can briefly address this potential application in the article.

Regarding to the microfluidic device application on PS beads, please refer to the following manuscript.

[1] Y.-C. Kung, K.-W. Huang, W. Chong, and P.-Y. Chiou, “Tunnel Dielectrophoresis for Tunable, Single-Stream Cell Focusing in Physiological Buffers in High-Speed Microfluidic Flows”, Small, Vol. 12(32), pp. 4343-4348, 2016

[2] Y.-C. Kung, K.-W. Huang, Y.-J. Fan, and P.-Y. Chiou, “Fabrication of 3D High Aspect Ratio PDMS Microfluidic Networks with A Hybrid Stamp”, Lab on a Chip, Vol. 15(8), pp. 1861 – 1868, 2015.

In general, this manuscript is well prepared and all the data supported the proposed preparation methods.

Author Response

Thank you for your suggestion. The relevant description has been added to the revised manuscript: “In addition, PS nanoparticles with negative charge can also be used for calibration of some microfluidic devices because of their uniform and controllable particle size. In microfluidic applications such as flow cytometry, cell sorting and imaging, it is a key capability to three-dimensionally focus randomly distributed particles into a single flow [14,15].” The references you provided have also been added:

  1. Kung, Y.C.; Huang, K.W.; Fan, Y.J.; Chiou, P.Y. Fabrication of 3D high aspect ratio PDMS microfluidic networks with a hybrid stamp. Lab on a Chip. 2015, 15, 1861-1868, doi:10.1039/C4LC01211A.
  2. Kung, Y.C.; Huang, K.W.; Chong, W.; Chiou, P.Y. Tunnel Dielectrophoresis for Tunable, Single‐Stream Cell Focusing in Physiological Buffers in High‐Speed Microfluidic Flows. Small.2016, 12, 4343-4348, doi: 10.1002/small.201600996.

Reviewer 2 Report

 In this paper, the author synthesis a uniform PS microspheres with different size by solvothermal method. Blend films with homogeneously dispersed PS microspheres in PDMS/P (VDF TrFE) were prepared. The influences of the mass fraction and the size of PS  microspheres on the melting and crystallization characteristics, the surface morphology and the dielectric hydrophobicity of the composite films and the contact angle were systematically analyzed.

Thank you for the information which you discussed clearly during the results.

In table 1 The Tg, may you discuss in details the changes occurs in the Tg

In page 7 line 221: table 4.1 or table 1

The references are too much, decrease them

The paper written well, I accept the paper after a few corrections

Author Response

Question 1: In table 1 The Tg, may you discuss in details the changes occurs in the Tg.

Answer 1: Thank you for your insightful suggestion. The description has been modified in the manuscript: “Compared with the Tg of pure PDMS/P(VDF-TrFE) organic matrix blend film, the Tg value first increases and then decreases with the increase of mass fraction of PS. This is mainly because the Tg of PS is around 110℃, and PS make the Tg of blend films move towards the higher temperature direction. As shown in Figure 5, the ratio of amorphous composition to crystalline decreases with increasing the PS content, which leads to the lower temperature (Tg) of the amorphous composition changing from glassy state to high elastic state. For the samples with the same PS doping content, the Tg value increases with increasing the particle size. This is mainly attributed to the fact that the crystallization peak strength decreases with increasing the particle size of PS microspheres, which is proved by the XRD of Figure 5.”

Question 2: In page 7 line 221: table 4.1 or table 1

Answer 2: Thank you for your kind reminder. “Table 1” has been confirmed in the manuscript.

Question 3: The references are too much, decrease them

Answer 3: Thank you for your professional advice. References have been reduced to 39 in the manuscript.